# The Role of Adhesion Molecules and Extracellular Vesicles in an In Vitro Model of the Blood–Brain Barrier for Metastatic Disease

**DOI:** 10.3390/cancers15113045

**Published:** 2023-06-03

**Authors:** Chiara Vasco, Ambra Rizzo, Chiara Cordiglieri, Elena Corsini, Emanuela Maderna, Emilio Ciusani, Andrea Salmaggi

**Affiliations:** 1Laboratory of Clinical Chemistry SMeL122, Fondazione IRCCS Istituto Neurologico “Carlo Besta”, 20133 Milan, Italy; vasco@ingm.org (C.V.); ambra.rizzo@istituto-besta.it (A.R.); elena.corsini@istituto-besta.it (E.C.); 2Preclinical Neuroimmunology Lab, Neurology IV Fondazione IRCCS Istituto Neurologico “Carlo Besta”, 20133 Milan, Italy; cordiglieri@ingm.org; 3Imaging Facility, National Institute of Molucular Genetics (INGM) “Romeo ed Enrica Invernizzi”, c/o Policlinico di Milano Hospital, Padiglione Invernizzi, Via Francesco Sforza 35, 20122 Milan, Italy; 4Neurology 5/Neuropathology Unit, Fondazione IRCCS Istituto Neurologico C. Besta, 20133 Milan, Italy; emanuela.maderna@istituto-besta.it; 5Neuroscience Department-Neurology/Stroke Unit, Ospedale A. Manzoni, ASST Lecco, 23900 Lecco, Italy; a.salmaggi@asst-lecco.it

**Keywords:** brain metastasis, brain metastasis molecular markers

## Abstract

**Simple Summary:**

Metastasis occurs when a tumor spreads from its original site to another part of the body. Several factors are involved in this complex process, including proteins that regulate adhesion between circulating tumor cells and target organs (called adhesion molecules). Additionally, it is shown in the literature that many tumors release small vesicles in the plasma that may contribute to metastatization. In this study, the authors focused on brain metastasis using an in vitro model of the physiological barrier that protects the brain, essentially consisting of a co-culture of endothelial cells and astrocytes. They examined the ability of breast and lung tumor cells to cross this barrier, as well as the expression of adhesion molecules. Furthermore, they collected and analyzed the vesicles released by tumor cells to determine their ability to create ruptures in the barrier. The data indicated that cells with intermediate levels of adhesion molecule expression were most effective in migrating through the barrier. Additionally, the vesicles produced by tumor cells were able to induce the death of endothelial cells, the primary cellular component of the barrier. Data reported support the role of adhesion molecules and tumor-produced vesicles in brain metastasis, in addition to other factors.

**Abstract:**

Metastatic brain disease (MBD) has seen major advances in clinical management, focal radiation therapy approaches and knowledge of biological factors leading to improved prognosis. Extracellular vesicles (EVs) have been found to play a role in tumor cross-talk with the target organ, contributing to the formation of a premetastatic niche. Human lung and breast cancer cell lines were characterized for adhesion molecule expression and used to evaluate their migration ability in an in vitro model. Conditioned culture media and isolated EVs, characterized by super resolution and electron microscopy, were tested to evaluate their pro-apoptotic properties on human umbilical vein endothelial cells (HUVECs) and human cerebral microvascular endothelial cells (HCMEC/D3) by annexin V binding assay. Our data showed a direct correlation between expression of ICAM1, ICAM2, β3-integrin and α2-integrin and the ability to firmly adhere to the blood–brain barrier (BBB) model, whereas the same molecules were down-regulated at a later step. Extracellular vesicles released by tumor cell lines were shown to be able to induce apoptosis in HUVEC while brain endothelial cells showed to be more resistant.

## 1. Introduction

Metastatic brain disease (MBD) has seen major advances both from a clinical and a biological standpoint in the last decade. Clinically, development of new predictive scores which take into account a number of clinical, radiological and biological factors has enabled confirming the existence of selected subgroups of patients that have a better prognosis than the historically identified three classes in the recursive partitioning analysis era. These three classes identified the best-surviving patients with a life expectation of 7 months only [1]. Clinical advances, which include the development of radiosurgery and of targeted therapies, have enabled achieving better results in survival, while a refinement in prognostic algorithms has first led to Sperduto’s 2012 disease-specific graded prognostic assessment tool, and more recently to the incorporation of molecular biology features in this tool [2], identifying subsets of patients with median survival of up to 46 months [3]. On the other hand, population studies show that survival is—as expected—far shorter in the context of real life, especially in elderly people [4]. Advances in overall survival in patients with MBD in the last decade most likely depend on a combination of factors, among which are the identification of targetable mutations in an increasing proportion of patients (for instance, EGFR, ALK, and ROS1 mutations which can be targeted by tyrosin-kinase inhibitors), the increasing use of immune checkpoint inhibitors leading to enhanced anti-tumor response by the host (anti-CTLA4, anti-PD1 and PD-L1 immunotherapies), and finally the widespread availability and increasing use of stereotactic radiosurgery [5]. Identification of targetable mutations in the context of MBD has pushed the balance towards a more aggressive attitude vis à vis the relevance of histological and molecular analysis of brain metastases. Although preliminary evidence seems to suggest that multiple brain metastases share the same biological profile in a single patient, it is not uncommon for brain metastases to exhibit a different molecular profile when compared to the primary tumor [6,7]. This emphasizes the renewed tendency of bioptic procedures in MBD, with the aim to ascertain the molecular biology profile of these lesions and allow optimization of therapy. At the same time, many efforts have been made to obtain the same biological information from less invasive procedures, such as liquid biopsy [8].

The development of a secondary localization of a solid tumor in another organ must necessarily involve the interaction between two key players: the tumor cell detached from the primary tumor mass dropped into the bloodstream and the endothelial cell which forms the inner lining of the blood vessels in the target organ, accomplishing a number of functions, among which is building a barrier between blood and tissue [9,10]. In particular, when metastatization occurs to the brain, metastatic cells must first be able to cross the intact blood–brain barrier (BBB): this process is not still completely clear because of the many actors involved, both on the surface of the metastatizing tumor cell and at the BBB itself [11], including the surrounding brain microenvironment.

In recent years, an emerging role for extracellular vesicles (EVs) in the biology of tumor growth and cross-talk with the target organ has been detected [9,12]. Tumor-released EVs may contribute to the formation of the necessary microenvironment (premetastatic niche), which will promote local tumor cell survival and growth [13,14]; indeed, the transfer of mRNA and microRNA mediated by EVs is a mechanism of genetic exchange between the cells now established in the premetastatic niche [15,16].

In this work, we have assessed the differential ability of various human breast and lung cancer cell lines to migrate through an in vitro model of the blood–brain barrier and through an endothelial cell monolayer of different origin (umbilical vein and cerebral microvasculature), aiming to relate this ability to the expression of a number of surface antigens. In separate sets of experiments, EVs obtained from the conditioned medium of tumor cell lines were incubated with the endothelial cells and EV-induced apoptosis was assessed and compared with baseline and conditioned medium-induced apoptosis.

## 2. Material and Methods

### 2.1. Cell Lines, HUVECs and HCMEC/D3 Cultures

A549, H460 (human lung cancer cells), MDAMB 231, MDAMB 453, SKBR3 and MCF7 (breast cancer cell lines) were all obtained from ATCC. Cells were cultured in medium (Dulbecco’s Modified Eagle’s Medium (DMEM) or RPMI1640, Thermo Fisher Scientific, Waltham, MA, USA) supplemented with 10% foetal bovine serum (FBS) (Thermo Fisher Scientific) and 1% penicillin/streptomycin (Thermo Fisher Scientific) at 37 °C in humidified atmosphere with 5% CO_2_.

Commercially available human cerebral microvascular endothelial cells (HCMEC/D3) (Sigma-Aldrich, Milan. Italy) were used as a model for brain endothelial cells. HCMEC/D3 were cultured in EBM-2 (Endothelial Basal Medium, Lonza, Walkersville MD, USA) supplemented with FBS 5%, 5 µg/mL bFGF 1 ng/mL (Sigma-Aldrich) and HEPES 10 mM.

Human umbilical vein endothelial cell (HUVEC) primary cultures were isolated from healthy donors essentially as previously described [17] and used in experiments within the tenth passage. The purity of HUVEC cultures was typically higher than 95% as assessed by flow cytometry after CD31 staining (polyclonal phycoerythrin-conjugated goat anti-human CD31, BD Bioscience, Milan, Italy). HUVECs were cultured in complete EBM (Endothelial Basal Medium, Lonza, Walkersville, MD, USA) medium with additional growth factors and antibiotics (EGM SingleQuots, Lonza) and grown on Petri dishes coated with Collagen S type I (0.3 μg/mL).

### 2.2. The Blood–Brain Barrier and Migration Assay

A model of the blood–brain barrier (BBB) was established as previously described in Rizzo et al., 2015 [18]. Briefly, rat astrocytes and HUVECs were co-cultured on both sides of a Transwell porous polycarbonate membrane (8 μm pores) in a 24 multiwell plate (Corning) previously coated with collagen IV at a concentration of 33 μg/mL (mouse collagen, IV, BD Biosciences) to allow cells to adhere. Astrocytes and HUVECs that constitute the BBB model were then co-cultured in EBM complete medium for four days before being used in transmigration experiments. The characterization of this model is described elsewhere [18]; however, some specific details are summarized in Appendix A. Tumor cells were harvested and resuspended in the complete medium (DMEM or RPMI) without FBS, pre stained with CFDA (15 min at 37 °C) and seeded into the Transwell upper chamber. Tumor cells (10^5^ cells in 200 µL) were allowed to migrate for 24 h into the bottom chamber containing 0.5 mL DMEM medium added with 10% FBS in a humidified incubator at 37 °C in 5% CO_2_. Migrated cells were counted by light microscopy in the medium in the bottom chamber. Moreover, cells that had attached to the lower side of the in vitro BBB (trapped cells) were visualized by a fluorescent microscope and counted.

The population of migrated cells was expanded for 4 weeks for the A549, H460, MDAMB 231, and MCF7 cell lines before flow cytometric analysis of surface markers as described in the following section.

### 2.3. Cell Line Characterization by Flow Cytometry

Phenotypical characterization of each cell line was performed by multicolor flow cytometry. Cells were harvested when 70–80% confluent, washed with PBS and incubated with PE- or FITC-conjugated antibodies for 1 h at 4 °C according to the manufacturer’s recommendation and then analyzed. Cells were characterized with the following antibodies: anti-CD56-FITC, anti-CD49b-FITC, anti-CD106-FITC, anti-CD197-FITC, and anti-CD146-PE (Beckman Coulter) (all from BD Pharmingen, Franklin Lakes, NJ, USA); anti-ICAM1-PE and anti-ICAM2-PE (both from Biolegend, USA); anti-alphaV-beta3-FITC (eBioscience^TM^, San Diego, CA, USA). Isotype-matched non-reactive fluorochrome-conjugated antibodies were used as controls and quantitative analysis was performed using a Navios EX flow cytometer (Beckman Coulter, Brea, CA, USA) with software Navios (Beckman Coulter, CA, USA). In this study, at least 10,000 events were analyzed for each sample excluding non-viable cells based on forward scatter and side scatter parameters. The data are expressed as the ratio of mean fluorescence intensity (MFI) of each specific antibody and the relative isotype control. Values greater than 1 indicate expression of the specific marker. The above-mentioned surface markers were analyzed in separate experiments both in the original cell line and in the population of transmigrated cells.

### 2.4. EV Isolation

EVs were isolated using Total Exosome Isolation Reagent (from cell culture media) (Invitrogen, cat 4478359) according to the manufacturer’s instructions.

### 2.5. Characterization of EVs

#### 2.5.1. Super-Resolution Microscopy

EVs from MCF7 and MDAMB 453 were prepared from Wheat-germ-agglutinin (OregonGreen 488, Invitrogen, USA). They were seeded at a high concentration in super-resolution mounting medium (ProLongGlass, Invitrogen) over 1.5 thickness super-resolution cover-glasses for acquisition in both laser scanning confocal microscopy (LCSM) using a Nikon EZ-C1 instrument with multi-spectral head and argon Laser, equipped with a 1.40 NA 100x oil objective, and in super-resolution structure illumination microscopy (SIM) using the 2D SIM/TIRF modality with total internal reflection angle illumination using high power 488 laser and 2D-grid SIM detection followed by FFT Gustaffson algorithm reconstruction over a Nikon N-SIM/N-STORM instrument, equipped with 1.49 NA 100× TIRF oil objective. Reconstructed images were analyzed using NIS-Elements V.5.31 software (Lim/Nikon Instruments) for image quantification, using an ad hoc implemented pipeline of image processing and segmentation (GA3 module in NIS-Elements). Data were further elaborated with GraphPad PRISM v.9.

#### 2.5.2. Transmission Electron Microscopy

EV suspensions were fixed for 1 h with 2.5% glutaraldehyde in 75 mM cacodylate buffer also containing 2 mM MgCl_2_ and 2 mM NaCl. For transmission electron microscopy (TEM), 5 µL of the sample was applied to carbon-coated copper grids and negatively stained with 2% uranyl acetate. To perform scanning electron microscopy, the fixed samples were frozen to glass slides with liquid nitrogen, washed 4 times with buffer (5, 10, 30, and 60 min), postfixed with 1% aqueous osmium tetroxide for 1 h, and washed again twice with buffer (10 min and overnight) and 3 times with double-distilled water (5, 20, and 30 min). After dehydration with a graded acetone series, the samples were processed via critical point drying. The glass slides were mounted onto aluminum stubs and sputter-coated with platinum for 40 s. TEM was performed on a Zeiss EM 912 (Carl Zeiss AG, Oberkochen, Germany) at 80 kV. Images were acquired by using a Tröndle 2k × 2k slow-scan charge-coupled device camera (Tröndle Restlichtverstärkersysteme, Moorenweis, Germany).

#### 2.5.3. Flow Cytometry

Expression of EMMPRIN and adhesion molecules on EVs was performed by flow cytometry using anti-CD63-coated microbeads following the manufacturer’s protocol (Exosome-Human CD63 Isolation/Detection Reagent, Invitrogen). The following fluorochrome-conjugated antibodies were tested: anti-CD56-FITC, anti-CD49b-FITC, anti-CD106-FITC and anti-CD197-FITC (all from BD Pharmingen Franklin Lakes, NJ, USA); anti-ICAM1-PE and anti-ICAM2-PE (both from Biolegend, San Diego, CA, USA); anti-alphaVbeta3-FITC and anti-CD147-FITC (both from eBioscienceTM). The binding of a specific antibody was evaluated using the ratio between the MFI of beads-bound EVs incubated with isotype-matched non-reactive fluorochrome-conjugated antibodies and the MFI of the beads-bound EVs incubated with each specific antibody. Ratios greater than 1 indicate expression of the protein. Analysis was performed using a Navios EX flow cytometer (Beckman Coulter) with software Navios (Beckman Coulter).

### 2.6. Quantitative RT-PCR (qRT-PCR)

EVs were lysed in Lysis Binding Buffer and mRNA was isolated using Dynabeads^®^ mRNA DIRECT™ Purification Kit (Invitrogen). The isolated mRNA was reverse transcribed using iScript cDNA Synthesis Kit (Bio-Rad, Segrate, Italy). QRT-PCRs were performed with specific assays for caspase 3, caspase 8 and beta-actin (Applied Biosystems, Carlsbad, CA, USA) following the manufacturer’s protocol. Data were analyzed on ABI 7500 (Applied Biosystems, Carlsbad, CA, USA).

### 2.7. Conditioned Culture Media Collection

EV-free complete culture media were obtained removing any debris or EVs by ultracentrifugation of complete medium as described previously [15]. Each cell line was cultured in complete medium until sub-confluent. Complete medium was then replaced with EV-free complete medium, left in culture for additional 24/48 h at 37 °C in humidified atmosphere with 5% CO_2_, collected and stored at −80 °C.

To evaluate the pro-apoptotic properties of conditioned media and EV on HUVECs and HCMEC/D3, EVs were isolated from 1 mL of conditioned culture media by Total Exosome Isolation Reagent (cat. N. 4478359, Invitrogen, Waltham, MA, USA), following the manufacturer’s instructions and resuspended in 250 µL fresh EV-free medium to be used in the experiments.

### 2.8. Evaluation of Conditioned Medium/EV-Induced Apoptosis in Endothelial Cells

Sub-confluent HUVECs or HCMEC/D3 were harvested, resuspended in complete EBM medium and seeded in 12 wells culture plates (5 × 10^4^ cells/well) 24 h before the experiment. The following day, culture medium was replaced with 250 µL of conditioned medium or isolated EV from each of the cell lines and incubated for further 24 h. Endothelial cells were then harvested, washed in PBS and apoptosis was evaluated by flow cytometry using the Annexin V binding assay following the manufacturer’s protocol (Annexin V Apoptosis Detection Kits Cat. N. BMS500FI-20, eBioscience™). As control, HUVECs were incubated with EV-free EBM medium. Data are expressed as the difference between the percentage of apoptotic cells detected in each supernatant/EV and that detected in control wells.

### 2.9. Statistics

Statistical analyses were performed by Pearson’s correlation coefficient for correlations and by the Student-*t* test for adhesion molecule expression. Unless differently specified, the level of significance was set at *p*  <  0.05.

## 3. Results

### 3.1. Migration

The migration ability of human tumor cell lines was evaluated in a transmigration assay both counting the cells in the culture media in the bottom side of a Transwell (Figure 1A) and counting the number of cells that were able to transmigrate through the polycarbonate insert but that remained adherent to the barrier itself (Figure 1B). The frequency of cells found in the culture medium in the bottom side of the Transwell (migrated cells) was inversely related to the level of expression of adhesion molecules, especially ICAM1 and α2-integrin (Figure 1B), while the levels of expression of the same molecules were directly related to the number of “trapped cells” (Figure 1B). A trend of an inverse correlation between the percentage of migrated cells and the number of cells trapped in the insert was detected (Appendix A).

### 3.2. Expression of Adhesion Molecules

The expression of a number of adhesion molecules putatively involved in migration/transmigration was investigated in breast and lung tumor human cell lines. Figure 2 summarizes the results of this analysis, showing the data from the original cell lines and the subpopulation of transmigrated cells expanded from the bottom side of the Transwell (BBB-cell lines). ICAM1, ICAM2, α2 integrin, NCAM, and β3 integrin were all expressed at varying levels across the six cell lines, whereas MCAM and CCR7 were not clearly expressed in all cell lines (three out of six and two out of six, respectively).

When comparing the levels of expression of adhesion molecules between the original cell lines and the subpopulation of transmigrated cells expanded from those recovered in the bottom side of the Transwell, a significant decrease in expression of α2 integrin and ICAM1 was found in three out of four tumor-cell lines tested. This data, together with the trend of an inverse correlation between ICAM1 and alpha2-integrin and transmigration ability, suggests that these adhesion molecules may play a role in transmigration. In our experimental model, cells with higher levels of these adhesion molecules are more likely to remain in the lower side of the insert, rather than entering the bottom well of the Transwell.

### 3.3. Characterization of EVs Produced by Tumor Cell Lines

In our experimental conditions, we successfully isolated EVs from each of the in vitro cultured cell lines studied. We then used structure illumination super resolution microscopy (SIM) and TEM to further characterize EVs (Figure 3). The sizes of EVs from MCF7 cells and MDAMB 453 and A549 cells were found to be similar, with comparable areas, perimeters, and diameters (Figure 3B,C). These findings are consistent with those of other authors who used comparable techniques [19]. Flow cytometry analysis of EVs captured on anti-CD63-conjugated beads revealed the presence of EMMPRIN on all EV preparations and other adhesion molecules as summarized in Table 1.

### 3.4. Isolated EVs Induced Apoptosis of Endothelial Cells

It was hypothesized that molecules and/or soluble factors produced by tumor cells might be able to promote vascular permeability in other organs. To assess this, isolated EVs and conditioned media were used to measure their capability to induce apoptosis in endothelial cells. Our results showed that in vitro, HUVECs are typically 10–15-fold more susceptible to apoptosis than HCMEC/D3 (Figure 4A,B). In HCMEC/D3, supernatants from SKBR3 and MDAMB 231 produced only a minimal level of apoptosis, compared to the effect of EVs which seem to be even lower (Figure 4B).

In the SKBR3, MCF7, MDAMB 453 (breast tumor cell lines) and A549 (lung tumor cell line) cell systems, isolated EVs induced significantly higher levels of apoptosis in HUVECs than their respective supernatants. However, MDAMB 231 and H460 cells demonstrated a greater response to the conditioned media than to the isolated EVs (Figure 4A). A dose-dependent effect was also observed with increasing amount of EVs inducing higher levels of apoptosis in HUVECs. Since the absolute amount of EVs produced by each cell line was not quantified, we were unable to compare the pro-apoptotic effect of SKBR3-derived EVs with that of MCF7-derived EVs; instead, we only measured the effects of the varying concentration of EVs in each cell line (Figure 4C). RT-PCR analysis revealed the presence of RNA coding for caspase 3 and caspase 8 in the majority of EVs derived from the tumor cell lines (Figure 4D).

## 4. Discussion

Progress in the field of MBD has been relevant in recent years. Brain metastases are a frequent complication of cancer (20% of cases), especially for lung, breast and melanoma primary tumors [20,21,22,23]. However, major gaps persist in our knowledge about the development and growth of brain metastases, especially due to heterogeneity of original primary cancers [23]. These gaps stem from a number of factors, among which is the present lack of recommendations for MBD screening in a number of primary cancers, probably leading to gross underestimation of the prevalence of the disease [22]. Moreover, the BBB results in a challenge for MBD treatment because of the difficulty of drug delivery in a such a protected district [11]. Despite the difficulty of drug delivery through the BBB, in recent years an increasing availability of treatments for both primary and brain metastatic cancers has occurred: for this reason, a precocious finding of MBD might be able to influence disease progression and overall survival [22].

The emerging evidence of possible differences In the molecular profile of brain metastases as compared with primary cancer has further stressed the need for histological assessment of the brain lesions or alternatively—due to the anatomical location of the metastasis and to the invasiveness of the bioptic procedure—for surrogate imaging or biological markers for brain metastases and their biological features [24]. In this area, many biological parameters are the subject of active investigation, among which are circulating tumor cells, liquid biopsies, tumor DNA characterization and mutations analysis, and circulating extracellular vesicles [24,25].

These biological markers should be studied to determine the mechanisms underlying dissemination, for diagnostic and/or prognostic tools and finally as a therapeutic target.

Although the mechanisms by which systemic tumor cells penetrate the brain crossing the BBB are not completely known, some evidence showed that adhesion molecules are involved in circulating tumor cells/endothelial cells interaction [26]. Tumor cells may directly interact with brain microvascular endothelial cells increasing the permeability of the BBB or work “at distance” producing soluble factors and/or vesicles able to influence the microenvironment leading to brain invasion.

In our experiments, we found that the BBB model is much more efficient in blocking cancer cells transmigration than the HCMEC/D3 monolayer. For the SKBR3 and MCF 7 cell lines, less than 0.05% of cells are able to cross the BBB while a frequency of 15 to 30% are able to pass the monolayer of HCMEC/D3. Additionally, all the other cancer cell lines display a transmigrating efficiency under the threshold of 0.05%. These findings met our expectations reflecting the fact that endothelial barrier functions vary through organs: the blood–brain barrier offers an example of the tightest and efficient architecture in protection of the CNS through which only small and selected molecules or selected immune cells are able to transmigrate [10]. The simple endothelial barrier offers, instead, different levels of permeability from arterioles (small permeability) to venules (great permeability). α2-integrin, together with β1-integrin is a cell surface receptor for laminin, collagen, fibronectin and E-cadherin and plays a pivotal role in cell to cell or cell to extracellular matrix (ECM) interaction, in cell signaling and immune process. Tumor cells express α2integrin on their surface and this protein is able to modulate their functions, including apoptosis, cell motility, invasion and angiogenesis. It is known that the interaction of α2-integrin and its ligand may facilitate adhesion of tumor cells to the vascular endothelium and promote metastatization although recent evidence suggest that its downregulation at a later step of the metastatic process might be related to a more aggressive behavior. These latter data refer to bone metastases [27].

In our experimental conditions, we found a correlation between the tumor cell ability to tightly interact (trapped cells) with the endothelial layer and the expression of selected adhesion molecules; in particular, α2-integrin, ICAM1, ICAM2 and β3-integrin expression levels correlated with the adhesion phase of the migration process (see also Appendix A).

This observation was not surprising and confirms previously published data, although the data in the literature are not uniformly demonstrative of a straightforward relationship between α2integrin expression and increased metastatization [18,28,29].

As a matter of fact, ICAM1 downregulation has been shown to decrease migration of the MCF7 cell line and its blockade has been reported to inhibit cancer cell migration in vitro [30,31]. All together these data confirm a central role for this adhesion molecule in different cancer type cell migration. 

On the other hand, a striking reduction in the expression of some adhesion molecules was detected in the tumor cells recovered from the bottom side of the Transwell, i.e., cell at a more advanced stage in the metastatic process. Although our in vitro model lacks the influence of the brain microenvironment, it might be speculated that the expression of adhesion molecules is modulated according to the phase of the biological process. However, much of the complexity of brain metastatization still remains to be defined and even more so for the mechanisms regulating dormancy and its suppression in the brain [32].

It is still debated whether persistent damage to the endothelium occurs in the process of brain metastatization. Most of the evidence points to a dynamic interrelationship between circulating tumor cells and/or tumor derived EVs and the target organ endothelium. However, apoptosis might occur in consideration of the stress impinged upon the endothelium, with microvascular proliferation [33,34].

The in vitro data presented in this paper suggest that tumor-derived EVs may indeed induce apoptosis of endothelial cells. Previously published papers already showed that the brain invasion by circulating tumor cells might also take place in part by inducing apoptosis of endothelial cells [9,35,36,37]. Our in vitro data suggest that even tumor-released EVs have this potential, possibly activating caspase transcription in endothelial cells and promoting then fenestration and extravasation of tumor cells. However, other mechanism than apoptosis might account for loss of BBB morphofunctional integrity in this context.

In our experimental model, HCMEC/D3 displayed increased resistance to EVs/conditioned media-induced apoptosis compared to HUVECs. This was somehow surprising since brain endothelial cells may undergo apoptosis using both the intrinsic and extrinsic pathways [38]. Despite brain endothelial cells expressing the whole repertoire of cell death receptors, they have been shown to be resistant to Fas and TRAIL receptor-mediated cell death [39]. 

Due to the specific features of HCMEC/D3, a known in vitro model for the cerebral microvascular endothelium, it can also be hypothesized that they are less prone to be penetrated by tumor-released vesicles (or any other small-volume particle) and therefore are less sensitive to apoptosis than HUVECs [40]. In fact, the permeability of HCMEC/D3 cells has been reported to be increased in stress conditions and pro-inflammatory cytokines and chemokines such as TNFα and CCL2 via signaling pathways as JNK, PKC or NFκB, which could be included in the EVs produced by tumor cell lines [40,41]. For example, a study by Kuroda et al. [42] recently identified possible receptors for the uptake of exosomes derived from SK-Mel-28 melanoma cells in human brain capillary endothelial cells (HCMEC/D3).

Our data taken together confirm that metastatization to the brain is a complex process that has to take into consideration many aspects such as the blood–brain barrier structure, possible action of molecules released by circulating cancer cells (EVs release or surface expression), dynamic crosstalk between BBB, pericytes and astrocytes, inflammation status, the microenvironment and cancer genetic heterogeneity [6,10,11,43].

## 5. Conclusions

Overall, the present data confirm the role of the investigated molecules and of EVs in the modulation of the early stages of metastatization in an in vitro model of the blood–brain barrier. Further studies are needed to elucidate the dynamic changes underlying metastatization to the brain, with a persistent focus on the microenvironment and on the mechanisms regulating dormancy of metastatic cells of progression to clinically relevant metastases.

## Figures and Tables

**Figure 1 cancers-15-03045-f001:**
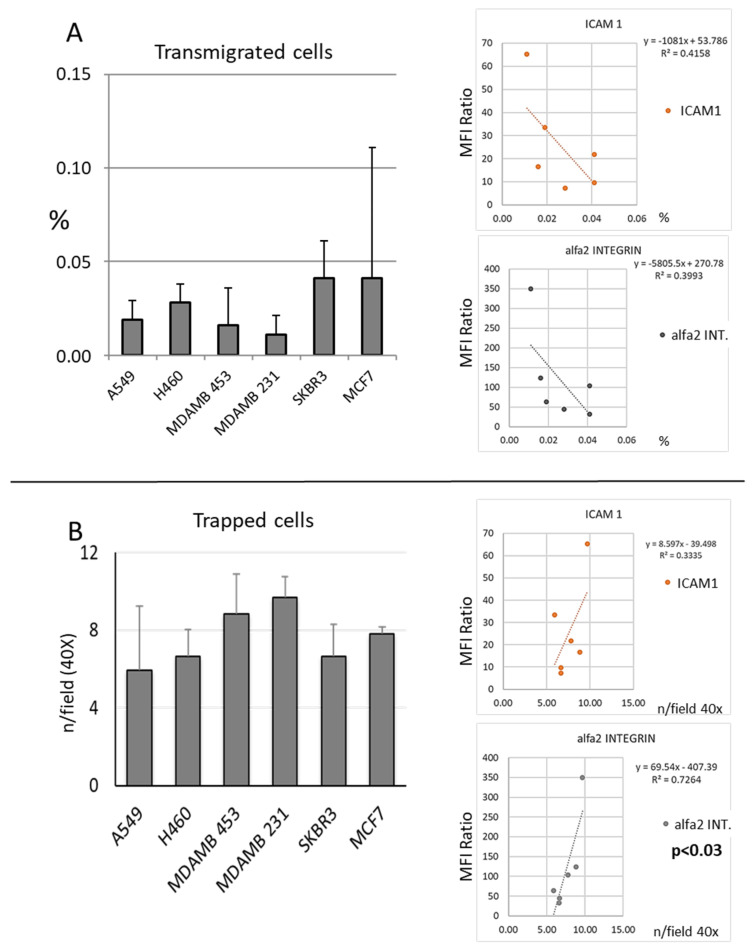
In vitro migration ability of breast and lung cancer cells lines. In panel (**A**) are summarized data regarding the proportion of cells counted in the lower well of the Transwell chamber expressed as % of cell transmigrated (see M&M). Although not significant, a trend of inverse correlation between transmigration ability and the rate of expression of adhesion molecules ICAM1 and a2 integrin in particular was detected. Cells attached to the lower side of the Transwell, i.e., cells which have been able to the cross endothelial cell monolayer-coated polycarbonate membrane, but that have remained attached to the lower face of the Transwell, are reported in panel (**B**). Higher adhesion molecule expression of the specific cell line directly correlated with the number of cells trapped (Appendix A). Data are expressed as n. of cells/field (40×). Correlations between migration parameters and ICAM1 or α2-integrin expression indexes are reported on the right side of panels (**A**,**B**). Data refer to at least two independent experiments. Error bars represent the standard deviations.

**Figure 2 cancers-15-03045-f002:**
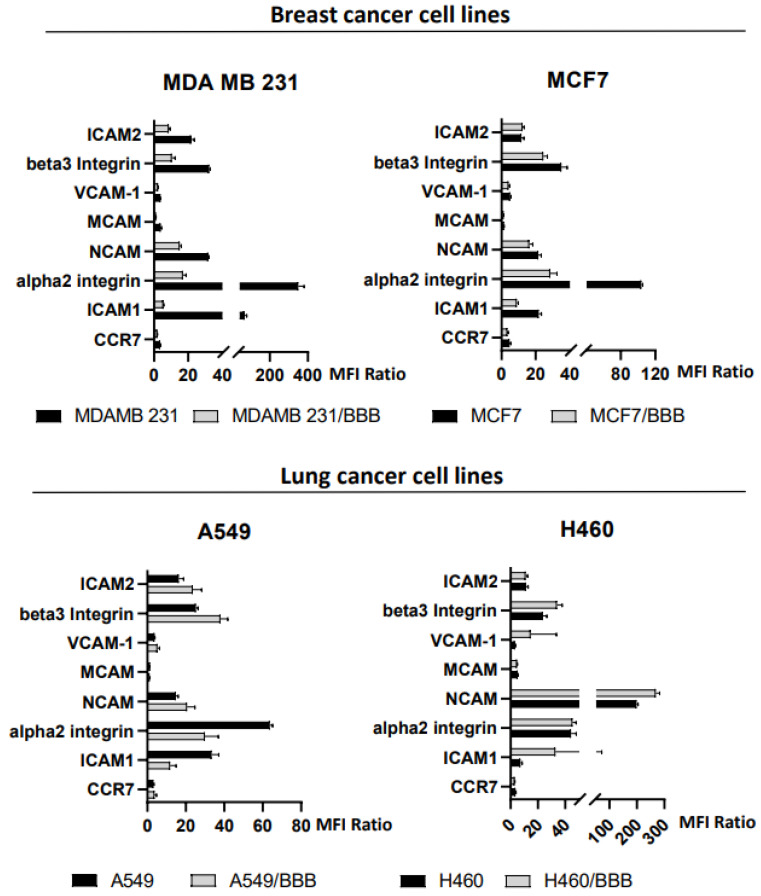
Expression of adhesion molecules in breast (upper panel) and lung (lower panel) human cancer cell lines as detected by flow cytometry, in original cell lines and their migrated subpopulation expanded in vitro (cell line/BBB, see M&M). A significant decrease in alpha2 integrin and ICAM1 expression was detected in transmigrated cells in three out of four cell lines. Data are expressed as the ratio of mean fluorescence intensity (MFI) of specific antibody and isotype control. Values higher than 1 indicate the expression of the molecule. Data refer to at least three independent experiments (see also Appendix A).

**Figure 3 cancers-15-03045-f003:**
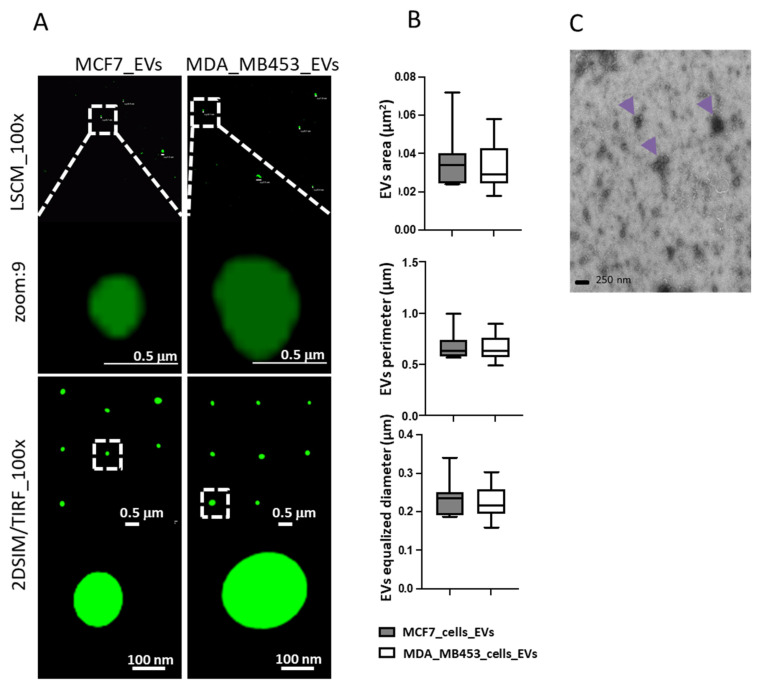
Characterization of EVs via advanced imaging, using both diffraction-limited microscopy (laser scanning confocal, LSCM, upper images in (**A**)) and structure illumination super resolution microscopy (SIM) in 2D-TIRF/SIM modality achieving 85 nm lateral resolution at 525 nm emission (lower images in (**A**)), to better resolve dimensions and morphology of EVs secreted by cells labelled with Wheat Germ Agglutinin 488, as to detect in green-emitting fluorescence the external EVs surface. Digital segmentation and quantification of SIM-acquired EVs images highlight similar dimensions between EVs from MCF7 cells and from MDAMB 453 cells, with comparable areas, perimeters and diameters (graphs in (**B**)). Panel (**C**) shows TEM performed on EVs isolated (arrows) from culture medium of A549.

**Figure 4 cancers-15-03045-f004:**
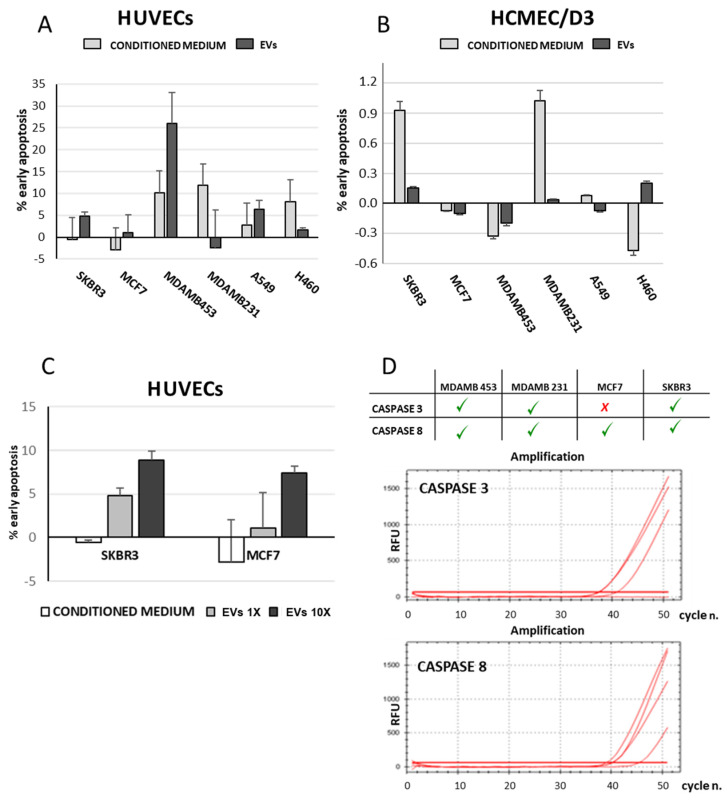
Apoptotic effect of conditioned media and isolated EVs on endothelial cells of different origin. Tumor cell line-derived EVs were added to culture media of HUVECs (panel (**A**)) or HCMEC/D3 (panel (**B**)) for 24 h. Endothelial cells were then harvested and apoptosis was measured by Annexin V binding assay. HUVECs showed to be more sensitive to apoptosis than HCMEC/D3. Moreover, EV-induced apoptosis of HUVECs increases with EVs concentration (panel (**C**)). Data are expressed as % of apoptotic cells and refers to at least two independent experiments. RT-PCR graphs reported in panel (**D**) show the presence of caspase 3 and 8 expression in EV-derived total RNA (RFU = relative fluorescence unit).

**Table 1 cancers-15-03045-t001:** anti-CD63-coated beads were used to analyze the presence of specific proteins on the surface of EVs isolated from culture media. All tumor cell line-derived EV preparations tested positive for EMMPRIN (first column left). With the exception of ICAM-2, each specific molecule was detectable on EVs in at least two cell lines. Expression or non-expression of the protein are reported with a “+” or a “–”, respectively (N.T.: not tested).

	EMMPRIN	ICAM 2	Beta 3 Integrin	VCAM	Alpha 2 Integrin	ICAM 1	CCR7	NCAM
A549	+	N.T.	N.T.	+	N.T.	+	+	+
MCF7	+	−	+	+	+	+	+	+
MDAMB 453	+	−	+	−	+	+	−	−
MDAMB 231	+	−	N.T.	−	N.T.	−	−	−
H460	+	+	N.T.	+	N.T.	+	+	−
SKBR3	+	N.T.	N.T.	+	N.T.	−	−	−

## Data Availability

The data that support the findings of this study are available from the corresponding author upon request.

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
