# Peer review of "The Role of Adhesion Molecules and Extracellular Vesicles in an In Vitro Model of the Blood–Brain Barrier for Metastatic Disease"

_cancers, 2023, doi:10.3390/cancers15113045_

Round 1

Reviewer 1 Report

The authors propose an original study based on transmigration of cancer cells across the blood-brain barrier (BBB) helped by a pro-apoptotic effect induced by their EVs on BBB endothelial cells. 

The transmigration part is clear and well-detailed based on a study of adhesion molecules on cancer cell lines used. However, some major points need to be considered and exploited to fulfill this proposed article:

- The in vitro BBB model used need to be checked for permeability, TEER and/or expression of tight junctions or tight junction-associated proteins to better clarity its integrity and a so-called barrier property. This remark is available for HCMEC/D3 and HUVEC models.

- Despite a nice approach by super resolution microscopy, the characterization of the EVs used (more related to medium EVs than small or large EVs) needs to be deepened in agreement with the Minimum Information for the Study of Extracellular Vesicles (MISEV) standards to confirm at least  their integrity and some key proteins carried by EVs.

Author Response

- The in vitro BBB model used need to be checked for permeability, TEER and/or expression of tight junctions...

Transmigration experiments have been performed using a previously characterized model published as stated in M&M (Rizzo et al. 2015). In this model, ultrastructural analysis by electron and confocal microscopy indeed allowed to detect the presence of tight junction as well as expression of occluding, Zo-1 and vinculin by endothelial cells and permeability to L-glucose. We now provide documentation of this in supplementary figure 1.

- Despite a nice approach by super resolution microscopy, the characterization of the EVs used (more related to medium EVs than small or large EVs) needs to be deepened...

We apologize since the first paragraph of 2.4 EVs isolation, should not have been included in the final version of the manuscript. As a matter of fact, all experiments concerning EVs reported in the manuscript have been performed with EVs isolated by Total Exosome Isolation Reagent.

Although not extensive, a characterization of Total Exosome Isolation Reagent-obtained EVs has indeed been performed and we provide it in the revised version of the manuscript (see table 1 and Fig. 3 Panel C).

Reviewer 2 Report

I was looking forward to review the manuscript by Vasco et al. as the title sounded highly interesting to me.

Unfortunately the high expectations I had from the title did not align with the scientific quality of the presented work.

I have two fundamental criticisms which are not allowing to accept this manuscript for publication:

1) The isolated EVs have not been characterized by any(!!) of the ISEV-recommended methods. Neither NTA, DLS, Western blot, or TEM have been performed by the authors. This is a no-go for EV-related research. On top, the authors used different methods (UC vs. precipitation kit) to isolate the EVs in different experiments. Where is the quantification of the vesicle number (concentration) applied? Different cell lines release very different amounts of vesicles...

2) Similar is true for the applied cell models, especially the hCMEC/D3 cells: No TEER value, IF staining of typical markers, no FITC-dextran permeability was measured, to proof that the cell layer formed a tight barrier. To be honest, I also highly doubt that hCMEC/D3 will form any relevant barrier on inserts with 8µm pores...

All these points make it impossible to draw reasonable conclusions from the subsequently performed experiments.

I'm sorry to say it that harsh: In my opinion, this whole study has to be repeated from scratch, starting with a proper, ISEV-compliant characterization of the used EVs + proper setup and use of appropriate controls for the used BBB models. To use HUVECs in a BBB model is anyhow very questionable as well...

Author Response

  • The isolated EVs have not been characterized by any(!!) of the ISEV-recommended methods. Neither NTA, DLS, Western blot, or TEM have been performed by the authors. This is a no-go for EV-related research...

We thank the reviewer for the observation. We apologize since the first paragraph of 2.4 EVs isolation, should not have been included in the final version of the manuscript. As a matter of fact, all experiments concerning EVs reported in the manuscript have been performed with EVs isolated by Total Exosome Isolation Reagent.

Although not extensive, a partial characterisation of Total Exosome Isolation Reagent-obtained EVs has indeed been performed and we provide it in Table 1 and figure 3 (panel C) in the revised version of the manuscript.

  • Similar is true for the applied cell models, especially the hCMEC/D3 cells: No TEER value..

All transmigration experiments reported in the manuscript have been performed using a characterized model previously published as stated in M&M (Rizzo et al. 2015). This model, consisting of a co-culture of HUVECs and rat astrocytes, was shown to have evidence of tight junction formation and expression of occludin, Zo-1 and vinculin proteins by endothelial cells via ultrastructural analysis by electron and confocal microscopy. We have not included the characterization data in the manuscript as it does not fall within the scope of the study. Anyhow, we thank the reviewer for the suggestion and confirmation data are provided as supplementary figure (S1) in the revised version of the manuscript.

hCMEC/D3 cells have been used in transmigration experiments only to compare the results with those obtained with the BBB model by Rizzo et al. and the results are not reported in the manuscript, but hinted in discussion. As a matter of fact, this was not clearly stated in the manuscript and we apologize for that.

We must respectfully disagree as far as the considerations on the BBB model are concerned. In our opinion, it is not at all surprising that endothelial cells, grown in close contact with astrocytes, do develop morpho-functional features of brain microvascular endothelium. Although we have limited experience with hCMEC/D3, if HUVECs are capable of forming tight junctions and other BBB features in some conditions, we would expect hCMEC/D3 cells to do the same.

Reviewer 3 Report

In this manuscript, Vasco and colleagues investigated the role of EVs and adhesion molecules on brain metastases of lung and breast cancer cells. Overall, I find the data very poorly presented. The results do not support the claims and the analysis is incomplete. There is no consistency between the results to support the model proposed by the authors. As a consequence, I reject this manuscript for publication.

Some recommendations to improve the manuscript are listed below.

1)      Abbreviations should be defined when used for the first time in the abstract and introduction (ex. EVs).

2)      Several grammatical errors. Several sentences are too long and poorly structured making it difficult to understand. I would recommend reviewing the manuscript carefully and modifying as needed.

Line 40 – typo ‘’uo tp’’ should be up to.

Line 52 – single patients

Line 194 – typo MIgration

Line 239 – BBB-cell lines, the cells tested here are not BBB cell lines but rather lung or breast cancer lines. This should be modified.

3)      If the authors want to claim that they are using a BBB model, they should validate the model with TEER values and expression of key molecules of the BBB such as tight junction proteins.

4)      Figure 1 needs to modified. The resolution is extremely poor, axes are poorly labeled and font size is not consistent between panels. In addition, the data does not support the claims. The R-square values point to no correlation in the majority of cases.

5)      Figure 2 – why were only 4 of the 6 cell lines used in Figure 1 included here? I would recommend including all 6 in all figures or reducing to 4 cell lines in all figures for consistency.

6)      Figure 3 – even though the authors could decide to show the results for only 2 cell lines, the results from other cell lines should be discussed in the results section of this figure. Was there a correlation between all cell lines? What does it imply to have similar MVs in all cell lines or not? I don’t see the point of this figure without such explanation of the results.

7)      Figure 4 investigates the effect of MVs and factors found in the supernatants on vascular permeability yet the results only present the level of apoptosis under the respective conditions. Several other factors could contribute to lower vascular permeability such as lower expression of tight junction proteins and should be at least discussed in the context of this figure. There is no real trend that supports the model proposed by the authors in this figure. Again, very poor resolution and labeling. The fonts are so small in panel D that it is impossible to read.

8)      In general, I find the quality of the figures very poor. The resolution is poor, the labeling is incomplete and confusing and the fonts are not consistent between figures and panels.

Author Response

Abbreviations should be defined ...

Several grammatical errors. Several sentences are too long and poorly structured making it difficult to understand ...

We have corrected the inconsistencies detected by the reviewer, thank you.

If the authors want to claim that they are using a BBB model, they should validate the model ...

All transmigration experiments reported in the manuscript have been performed using a characterized model previously published as stated in M&M (Rizzo et al. 2015). This model, consisting of a co-culture of HUVECs and rat astrocytes, was shown to have evidence of tight junction formation and expression of occludin, Zo-1 and vinculin proteins by endothelial cells via ultrastructural analysis by electron and confocal microscopy. We have not included the characterization data in the manuscript as it does not fall within the scope of the study. Anyhow, we thank the reviewer for the suggestion and confirmation data are provided as supplementary figure (S1) in the revised version of the manuscript.

Figure 1 needs to modified ...

Resolution of the figures have been improved. We agree that R squared values point to no statistically significant correlation in the majority of cases as reported in supplementary figure 3.

Figure 2 – why were only 4 of the 6 cell lines used in Figure 1 included here? ...

In SKBR3 and MDAMB 453 cell lines, we were not able to expand transmigrated cells therefore no data on adhesion molecules expression is available except their basal expression which is  reported in supplementary figure 4 of the revised version of the manuscript. However, we could isolate EVs by all cell lines that have been used to induce apoptosis in HUVECs and HCMECs and for evaluating the effect of EVs concentration on endothelial cells apoptosis. Removing any data related to these cell lines would significantly reduce the amount of data reported in the manuscript.

Figure 4 investigates the effect of MVs and factors found in the supernatants on vascular permeability yet the results only present the level of apoptosis under the respective conditions. Several other factors could contribute ...

We have added a short sentence in the discussion acknowledging that a variety of factors other than apoptosis may indeed be relevant in this context.

Figure 3 – even though the authors could decide to show the results for only 2 cell lines, the results from other cell lines should be discussed ...

The characterisation of EVs by super resolution microscopy was only performed to evaluate the quality of EVs obtained with the kit used in the manuscript. Since the method is complex and time consuming, this data is available only for the two cell lines shown.

Although not extensive, a partial characterisation of Total Exosome Isolation Reagent-obtained EVs has indeed been performed and we provide it in Table 1 and figure 3 (panel C) in the revised version of the manuscript.

8)      In general, I find the quality of the figures very poor ...

We have improved the resolution of the figures. However, we are concerned that the submission process of resizing the figures to be included in the Word document may affect the quality of the figure. We may provide high resolution figures.

Round 2

Reviewer 1 Report

The authors did a nice revision o the previous manuscript and took all my remarks into account to modify/add/change/argue to some key points regarding the BBB and EV characterization.

Further investigations could complete this proposed study, however I accept this revised version for publication. 

Author Response

Many thanks for your contribution and suggestions

Reviewer 2 Report

I appreciate the efforts of the authors to improve the manuscript und clarify some points in the text.

Nonetheless, all results are still based on assumptions which are not scientifically proven in the presented study. The use of one marker (CD63) in a bead-assay is insufficient to characterize EVs. At least three markers + a negative marker must be shown. This assay also does not answer the question how many EVs were present and used in the experiments, nor any reliable data on the size distribution of the EVs could be provided by the authors.

The authors also provided immunofluorescence and TEM images of their HUVEC-BBB model in a supplementary figure. The quality of the IF images is unfortunately very poor and do not support the authors' claim that Occludin or ZO1 are present (at least a Western Blot would be needed in addition). First of all, negative controls are missing (2nd ab only), secondly, both markers are mislocalized - they should be seen on lateral cell-cell borders, not overall the whole cell body. Therefore I highly doubt, that those are specific stainings, or, the cells do not form proper cell-cell junctions. Single TJs might of course be visible in TEM if one searches for.

Similar to the EV characterization, my criticism on the cell model methodology remains unchanged. The barrier integrity - which is absolutely essential for this study - was not proven by the authors. TEER, Dextran permeability measurements,... are still missing.

For a journal with an Impact Factor over 6 points, I must expect much higher data quality for a manuscript to be acceptable.

Author Response

I appreciate the efforts of the authors to improve the manuscript und clarify some points in the text.

Nonetheless, all results are still based on assumptions which are not scientifically proven in the presented study. The use of one marker (CD63) in a bead-assay is insufficient to characterize EVs. At least three markers + a negative marker must be shown. This assay also does not answer the question how many EVs were present and used in the experiments, nor any reliable data on the size distribution of the EVs could be provided by the authors.

We see the reviewer’s point. As far as EVs quantification we have done a number of experiments using DLS (Izone) at the beginning of the project not obtaining reliable results. However, although not exhaustive, the characterization of size distribution of EVs is reported in fig. 3.

The authors also provided immunofluorescence and TEM images of their HUVEC-BBB model in a supplementary figure. The quality of the IF images is unfortunately very poor and do not support the authors' claim that Occludin or ZO1 are present (at least a Western Blot would be needed in addition). First of all, negative controls are missing (2nd ab only), secondly, both markers are mislocalized - they should be seen on lateral cell-cell borders, not overall the whole cell body. Therefore I highly doubt, that those are specific stainings, or, the cells do not form proper cell-cell junctions. Single TJs might of course be visible in TEM if one searches for.

Negative controls (2nd Ab only) have been included in all IF experiments as normal laboratory praxis and the signal shown in the images reported in the figure is specific. We feel that taking images of a substantially black background would not add much to the manuscript. We provide new IF images for ZO-1 and Vinculin (sup. fig. 1).

Similar to the EV characterization, my criticism on the cell model methodology remains unchanged. The barrier integrity - which is absolutely essential for this study - was not proven by the authors. TEER, Dextran permeability measurements,... are still missing.

This BBB model has been extensively studied previously as stated in the manuscript (L-glucose permeability included). In the present work the integrity of the barrier was checked using thin layer microscopy (less time consuming compared to electron microscopy). For each transmigration experiment, we set up an artificial barrier in an extra transwell, which we used to verify the presence of both astrocytes and endothelial cells in a monolayer. We only accepted experiments if the two conditions were both satisfied. Supplementary Fig. 1 has been integrated with a new panel containing a brief explanation. In our experimental conditions, TEER measurements were inconsistent.

Reviewer 3 Report

Although I appreciate the efforts made by the authors to improve the initial version of this manuscript and address reviewer's comments, there are still important pitfalls to this study.

First, the fact that the BBB model has been characterized previously does not justify the lack of characterization in this study. At least the TEER values should be verified to confirm that the integrity of their model could be reproduced. The main conclusions of the authors are based on the presence of a tight barrier which is only an assumption in this case.

Although there is a trend towards an indirect correlation between the ability to transmigrate and the expression of adhesion molecules, it is not significant. The authors should review their wording when making their conclusions (see Line 281-282 for an example where the authors claim an inverse correlation).

The overall presentation of the figures should be improved further. Many font's are not visible, the resolution of the fonts is very poor in some cases, some axes are not labeled. In Figure 4 panel A, labeling overlaps with the graph whereas in panel B the labeling is below the graph.

Author Response

Although I appreciate the efforts made by the authors to improve the initial version of this manuscript and address reviewer's comments, there are still important pitfalls to this study.

First, the fact that the BBB model has been characterized previously does not justify the lack of characterization in this study. At least the TEER values should be verified to confirm that the integrity of their model could be reproduced. The main conclusions of the authors are based on the presence of a tight barrier which is only an assumption in this case.

Given our shared concern regarding the referee's comments, for each transmigration experiment, we set up a BBB in an extra transwell, which we observed via thin layer microscopy to verify the presence of both astrocytes and endothelial cells in a monolayer. We only accepted experiments if the two conditions were both satisfied. Supplementary Fig. 1 has been supplemented with a new panel containing a brief explanation. In our hands, unfortunately, TEER values were never reliable.

Although there is a trend towards an indirect correlation between the ability to transmigrate and the expression of adhesion molecules, it is not significant. The authors should review their wording when making their conclusions (see Line 281-282 for an example where the authors claim an inverse correlation).

The sentence has been modified accordingly.

The overall presentation of the figures should be improved further. Many font's are not visible, the resolution of the fonts is very poor in some cases, some axes are not labeled. In Figure 4 panel A, labeling overlaps with the graph whereas in panel B the labeling is below the graph.

We apologize for the missing information. Figures have been edited again according to the reviewer’s suggestions.

Round 3

Reviewer 2 Report

The authors achieved minor improvements in this manuscript version, but fundamental flaws in experiment design and EV methods remained untouched.

What if the observed effects and differences between EVs of different cell lines are influenced by vesicle numbers applied? The authors could not provided any information, how many EVs they have isolated and applied. Different cell lines release very different numbers of vesicles...

The fact that DLS results were "not reliable" in their own words, indicates that EV preparation methods would have had to be improved BEFORE starting a large set of experiments. Usage of advanced microscopy techniques does not compensate for measuring basic EV properties like size distribution and vesicle concentration.

As a side note: Precipitation kits are know to co-purify a vast number of other particles, protein aggregates etc. It should be clear, that the applied samples are a mixture of EVs plus other media components, etc.

The newly provided IF images (besides being presented with different magnification: Vinculin much smaller than ZO1...) do not prove a tight monolayer. When zooming in, it is very obvious that there are large gaps  (holes) between the cells.

Author Response

What if the observed effects and differences between EVs of different cell lines are influenced by vesicle numbers applied? The authors could not provided any information, how many EVs they have isolated and applied. Different cell lines release very different numbers of vesicles...

We agree with the reviewer’s observation. As a matter of fact, we never compared the effects of EVs from different cell lines; instead, we evaluated the effects of EVs vs. supernatants of the same cell line (Fig 4). When we compared the effects of EVs dose escalation, we considered the different concentrations, not different cell lines; however, to avoid any misunderstandings, we have added a brief sentence in the Results section to acknowledge this limitation of our study.

The fact that DLS results were "not reliable" in their own words, indicates that EV preparation methods would have had to be improved BEFORE starting a large set of experiments. Usage of advanced microscopy techniques does not compensate for measuring basic EV properties like size distribution and vesicle concentration.

As a side note: Precipitation kits are know to co-purify a vast number of other particles, protein aggregates etc. It should be clear, that the applied samples are a mixture of EVs plus other media components, etc.

Although we acknowledge the limitation of our study regarding the characterization of EVs, the reagent we used for their isolation has been extensively analyzed in other studies with successful results, suggesting that it is a viable alternative to ultracentrifugation. (Soares Martins T, PLoS ONE 13(6): 2018: e0198820; Helwa I et al. PLoS ONE 12(1): 2017; e0170628).

The newly provided IF images (besides being presented with different magnification: Vinculin much smaller than ZO1...) do not prove a tight monolayer. When zooming in, it is very obvious that there are large gaps  (holes) between the cells.

We took IF images in various experiments and different time points from the beginning of the co-colture of rat astrocytes and HUVECs. The localization of the vinculin and ZO-1 were most clearly detectable before the monolayer completely covered the polycarbonate insert. Vinculin and ZO-1 IF images have been taken after 60 hours after the beginning of the co-culture. This explains the presence of gaps between the cells but clearly shows the correct localization of the proteins. All this has been reported in the figure legend.

Reviewer 3 Report

Dear author, 

I appreciate your efforts to address previous comments. I would recommend that you go through the figures one more time to adjust font size, resolution and ensure consistency in the labeling and titles. 

Once this is done, I approve this manuscript for publication.

Author Response

I appreciate your efforts to address previous comments. I would recommend that you go through the figures one more time to adjust font size, resolution and ensure consistency in the labeling and titles.

Figures have been carefully checked and modified accordingly.